# Role of Oxidative Stress and Neuroinflammation in Attention-Deficit/Hyperactivity Disorder

**DOI:** 10.3390/antiox9111039

**Published:** 2020-10-23

**Authors:** Juan Carlos Corona

**Affiliations:** Laboratory of Neurosciences, Hospital Infantil de México Federico Gómez, Mexico City 06720, Mexico; jcorona@himfg.edu.mx; Tel.: +52-55-5228-9917

**Keywords:** oxidative stress, neuroinflammation, attention-deficit/hyperactivity disorder (ADHD), medications

## Abstract

Attention-deficit/hyperactivity disorder (ADHD) is a neurodevelopmental disorder of childhood. Although abnormalities in several brain regions and disturbances of the catecholaminergic pathway have been demonstrated, the pathophysiology of ADHD is not completely understood, but as a multifactorial disorder, has been associated with an increase in oxidative stress and neuroinflammation. This review presents an overview of factors that increase oxidative stress and neuroinflammation. The imbalance between oxidants and antioxidants and also the treatment with medications are two factors that can increase oxidative damage, whereas the comorbidity between ADHD and inflammatory disorders, altered immune response, genetic and environmental associations, and polymorphisms in inflammatory-related genes can increase neuroinflammation. Evidence of an association with these factors has become valuable for research on ADHD. Such evidence opens up new intervention routes for the use of natural products as antioxidants that could have potential as a treatment against oxidative stress and neuroinflammation in ADHD.

## 1. Introduction

### 1.1. Attention-Deficit/Hyperactivity Disorder

Attention-deficit/hyperactivity disorder (ADHD) is a prevalent neurodevelopmental disorder in children characterized by inattention, hyperactivity, and/or impulsivity [1,2,3] that impairs the psychological, social, academic, and occupational function [4]. The disorder is linked with a range of comorbidities (e.g., depression, anxiety, substance use disorders, conduct disorders, criminal behavior), developmental conditions (e.g., autism spectrum disorders), and physical conditions [5,6,7,8]. Clinical diagnosis of ADHD is based on the presence of six or more symptoms that include inattention and hyperactivity/impulsivity and, according to the criteria of the fifth edition of the *Diagnostic and Statistical Manual of Mental Disorders* (DSM-5), should be carried out when symptoms impair academic, occupational, and social behavior, the onset is before 12 years of age and the symptoms can be observed in multiple settings in the clinical interview, including the gestational, developmental, and family history [9]. The worldwide prevalence of ADHD in children and adolescents is 7.2%, with about 4% shown to persist in adults, and the diagnosis of ADHD is more frequent in boys than in girls [4,10,11,12,13]. Although ADHD is a childhood disorder, a large percentage of children continue to have symptoms in adolescent life, and approximately 45% have symptoms as adults [1,14,15].

### 1.2. Medications for ADHD

Medications used for ADHD are divided into psychostimulants and non-psychostimulants, which have diverse delivery systems, formulations, and pharmacokinetic actions. Educational and psychosocial approaches are also used as a treatment for ADHD [4,16]. Psychostimulants such as methylphenidate (MPH) and amphetamine are the first-line therapy for ADHD. MPH and amphetamine improve the symptoms by blocking presynaptic dopamine and norepinephrine transporters, thus increasing catecholaminergic transmission in the striatum, prefrontal cortex, and hippocampus [4,9,17]. Some side effects have been observed with psychostimulants, such as insomnia, appetite loss, headache, dry mouth, anxiety, and nausea [2,18]; long-term psychostimulants treatment also has an effect on growth (particularly weight and height velocity) and causes euphoric effects and cardiovascular events [19,20].

The second-line treatment is with non-psychostimulants such as atomoxetine (ATX), which is a selective norepinephrine transporter inhibitor, and selective α-2 adrenergic receptor agonists, namely clonidine and guanfacine [9,21,22]. Non-psychostimulant medications usually are given to patients who cannot tolerate the side-effects of psychostimulants and have a poor response. However, some side effects of non-psychostimulants have been linked with somnolence, nausea, vomiting, diarrhea, decreased appetite, dizziness, fatigue, and changes in cardiovascular parameters [2,23].

### 1.3. Etiology of ADHD

The heritability of ADHD is high, with a range of 70–80% [1,24]. Genome-wide association studies have identified approximately 22% of the heritability, associated with 12 genome-wide significant risk loci and also the enrichment of copy-number variants [25,26,27,28]. Prematurity/low birth weight and exposure to environmental toxins and pesticides have been highlighted as risk factors in ADHD [29,30]. Maternal exposure to tobacco and alcohol, nutritional deficiencies, viral infections, and obesity during pregnancy are also associated with ADHD but can also be associated with genetic factors [31].

### 1.4. Pathophysiology of ADHD

The pathophysiology related to ADHD remains unknown. However, ADHD is associated with abnormalities in the brain due to cognitive and functional deficits. Additionally, links between ADHD and the dopamine levels in specific brain regions have been found: patients with ADHD have attenuated dopaminergic activity [32,33]. In that sense, it has been suggested that deregulation in catecholaminergic neurotransmission is the cause of the disorder [34,35]. Furthermore, extensive data indicate the contribution of oxidative stress as a pathophysiological cause of ADHD [3,36,37]. Also, there is evidence indicating neuroinflammation as a possible factor in ADHD [3,38,39,40]. In this review, the role of oxidative stress and neuroinflammation as possible factors involved in the pathophysiology of ADHD is discussed, along with a possible link to the medications used for treatment that may increase these factors.

## 2. Role of Oxidative Stress

Oxidative stress is a state produced by an imbalance between antioxidants and oxidants in the cells. The imbalance occurs as a result of the inappropriate function of the antioxidant system or by an excessive level of reactive oxygen species (ROS). Oxidative stress in the brain can harm the integrity of neurons because the brain is rich in polyunsaturated fatty acids (PUFAs) that are highly susceptible to oxidation, producing ROS [41,42]. This condition could cause oxidative damage of neurons, which are rich in mitochondria; mitochondria can regulate the redox state, ion homeostasis, apoptosis, cell signaling and, as the powerhouse of the cell, produce ATP, thus they can generate ROS, causing bioenergetic disturbances that lead to cell death or several disorders, including neurodegenerative and psychiatric diseases [43,44,45].

Mitochondrial dysfunction, genetic, and environmental factors can generate oxidative stress, and inflammation is a neuroprotective response to diverse types of tissue damage. If the tissue is inflamed, it causes an increase of ROS, which can lead to cell death [46,47]. The oxidative stress modifies the inflammatory response, thus, when there is a redox balance, the inflammatory response is a defense mechanism; when there is a redox imbalance, the signaling pathways that modulate the immune system are altered, leading to dysregulation of the immune response [48]. In a chronic state of oxidative stress, proteins and lipids oxidize and the DNA is damaged. ROS could also lead to the activation of astrocytes and microglia. High concentrations of ROS could activate the high secretion of proinflammatory chemokines and cytokines and produce a vicious circle [47,48]. Therefore, oxidative stress and neuroinflammation are mechanisms that coexist and are interrelated. The pathophysiology of ADHD is associated with oxidative stress and neuroinflammation, due to the imbalance between oxidants and antioxidants, catecholaminergic dysregulation, medications used for treatment, genetic and environmental factors, and all those factors could be producing oxidative stress and neuroinflammation which further increases the symptoms and as a result, triggering a vicious circle.

### 2.1. Oxidative Stress and Oxidant Levels

Numerous studies have demonstrated elevated levels of oxidative stress in ADHD, increasing the evidence for oxidative stress being a pathophysiological factor. The measurement of higher rates of ethane levels, as a non-invasive measure of oxidative breakdown of n-3 PUFAs in patients with ADHD, was also demonstrated [49]. Furthermore, elevated levels of malondialdehyde (MDA) have been observed in children with ADHD [50,51]. A randomized, double-blind, placebo-controlled study found that children with ADHD had increased damage to DNA, measured with 8-oxo-7,8-dihydroguanine (8-oxoG) [52]. Moreover, the levels of MDA in plasma from children and adolescents with ADHD were significantly higher than the controls [53]. An increase in lipid peroxidation in pediatric ADHD patients was evaluated using acrolein-lysine in urine samples [54]. Also, changes in plasma levels of xanthine oxidase (XO) were significantly higher in patients with ADHD [55]. The levels of MDA and the DNA damage indicator 8-hydroxy-2′-deoxyguanosine (8-OHdG) were statistically lower in children with ADHD [56,57]. The total oxidative status (TOS) and oxidative stress index (OSI) were higher in patients with ADHD than controls [58]. Furthermore, in children and adolescents with ADHD, the TOS and OSI were significantly higher than in healthy controls [59]. Also, the TOS was high in plasma of children and adolescents with ADHD [60]. In a meta-analysis of ADHD patients, an increase in oxidative stress was found [37]. Moreover, TOS and OSI were increased in children with ADHD [61]. No significant differences were detected in serum TOS and OSI levels in adults with ADHD [62]. Also, in the spontaneously hypertensive rat (SHR), used as an animal model for ADHD, an increase in ROS production measured using 2′-7′-dichlorofluorescein diacetate (DCFH-DA) was demonstrated in the striatum, hippocampus, and cortex [63]. In contrast, it was demonstrated that MDA levels were not significantly different in children with ADHD [64]. Moreover, in children with ADHD, increases in plasma MDA and urinary 8-OHdG levels were found [65]. Finally, the levels of MDA and free sulphydryl groups in the spleen were higher in 5-week-old SHR than in control rats [66]. Recently, evaluation of serum levels of hydroperoxide, an oxidative stress marker, was shown to be higher in preschool children with ADHD [67].

### 2.2. Nitrosative Stress

An increase in nitrosative stress (i.e., nitric oxide (NO) levels) and an impaired balance of oxidants and antioxidants were observed in children with ADHD [68]. In the SHR, damaged non-selective attention was improved with the nitric oxide synthase (NOS) inhibitor l-nitro-arginine methyl ester (l-NAME) [69]. In contrast, reduced blood NO levels have been reported [70]. On the other hand, NO levels were significantly higher in ADHD [53,71]. Also, changes in plasma levels of NOS were significantly higher in patients with ADHD [55]. The outcomes of oxidative and nitrosative stress are summarized in Table 1.

### 2.3. Antioxidant Levels in ADHD

In a randomized, double-blind, placebo-controlled study, it was found that children with ADHD had decreased total antioxidant status (TAS) [52]. Also, the TAS in children with ADHD was low [82]. Moreover, in a randomized, double-blind controlled trial, higher concentrations of adrenaline and noradrenaline were found in the urine of ADHD patients, which correlated positively with the degree of hyperactivity, and they were associated with high levels of oxidized glutathione disulphide (GSSG), which is an important marker of exhaustion of the antioxidant glutathione (GSH) [83]. In plasma samples of patients with ADHD, the antioxidant enzyme activity of glutathione peroxidase (GPx) was significantly lower, the superoxide dismutase (SOD) activity was not significantly different between patients and controls, and the catalase (CAT) activity was higher than in the controls, but not statistically significant [53]. It was also demonstrated that the serum levels of SOD1 were significantly lower in children with ADHD [84]. Moreover, in plasma, the levels of SOD, glutathione-S-transferase (GST), GPx, and CAT were significantly lower in children with ADHD [85]. Furthermore, a decrease in the salivary total antioxidant activity was observed in children with ADHD [86]. It has been demonstrated that levels of the antioxidant enzymes GST and paraoxonase-1 (PON1) were significantly lower in plasma from patients with ADHD [55], however, the antioxidant PON1 and thiol levels were no different in children with ADHD [56]. In addition, a significant increase in the salivary thiol levels was observed; in contrast, ceruloplasmin, which is an important extracellular antioxidant, did not show any significant change, but magnesium levels were significantly decreased in children with ADHD [87]. The TAS was increased in patients with ADHD [58], but TAS levels in plasma tended to decrease in children and adolescents with ADHD, and antioxidant enzymes such as PON, stimulated PON, and arylesterase (ARE) showed no differences in activity. However, a significantly lower thiol enzyme activity was found in the plasma of children and adolescents with ADHD [60]. TAS was also significantly lower in children and adolescents with ADHD than in controls [59]. In children with ADHD, it was demonstrated that the TAS, PON1, and ARE activities were decreased [61]. In the SHR, decreased GPx activity was found in the prefrontal cortex, but there was no difference in the other regions or in GSH, SOD, and CAT activities between the SHR and controls [63]. In adults with ADHD, the homocysteine level was lower and the serum folate level was higher. However, no significant difference was detected in serum vitamin B12 and the TAS [62]. In contrast, erythrocyte GSH and plasma retinyl palmitate levels were higher in patients with ADHD than in controls [65]. The antioxidant levels of melatonin were high in serum from children with ADHD [68]. Finally, the total antioxidant capacity (TAC), CAT, and GSH were significantly lower in children with ADHD [64]. Some of the opposing results observed in the different studies could be explained by the different methodologies, the participant selection criteria and the analysis used. Thus, the extensive data propose that a decrease in antioxidants and an increase in both oxidative and nitrosative stress in ADHD could contribute to its pathophysiology [3,36,37]. The antioxidant outcomes of are summarized in Table 2.

As demonstrated in Table 1 and Table 2, data on oxidative, nitrosative stress, and antioxidant levels are inconsistent in patients with ADHD. Altogether, some differences have been observed, although the changes are controversial, suggesting that patients with ADHD have heterogeneity in the antioxidant production, but their response to oxidative and nitrosative stress is insufficient, leading to oxidative damage. Thus, studies carried out so far point out that ADHD is associated with increased oxidative stress. However, there continues to be inconsistency in findings, and this may at least be partly attributed to differences in participants examined, oxidative stress markers tested, and protocols and samples utilized to examine appropriate markers. It is also acceptable that oxidative stress is associated with some ADHD symptoms and/or subtypes, across gender and age, but this has not yet been enough explored due to a lack of appropriate research. Consequently, methodological differences might underlie contradictory results. Additional research is therefore required to help clarify the importance of oxidative stress in ADHD and its pertinence for the treatment and prevention of ADHD.

### 2.4. ADHD Medications and Oxidative Damage

It has been demonstrated that treatment with MPH increases the generation of oxidative stress; in the brain of young rats, chronic treatment with MPH increased oxidative stress as assessed by thiobarbituric acid reactive species (TBARS) and protein carbonyl formation [73]. Also, treatment with MPH in the striatum of young and adult rats increases DNA damage [74]. On the other hand, it was demonstrated that chronic exposure to MPH in the brain of young rats increases mitochondrial complexes [75]. Moreover, chronic or acute treatment with MPH altered the activity of SOD and CAT enzymes in the brain of young rats [88]. Acute administration of MPH in young rats increased the production of superoxide in submitochondrial particles in the cerebellum and hippocampus [76]. Additionally, in the prefrontal cortex of juvenile rats, chronic MPH treatment induced an increase in oxidative stress, protein damage, and lipid peroxidation [77]. Furthermore, an increase in oxidative stress was shown with acute and chronic MPH treatment in the SHR [78]. Finally, acute administration of high doses of MPH in adult rats produced oxidative damage, reduced GSH, SOD, GPx, and glutathione reductase (GR) activities, and provoked neurodegeneration in the cerebral cortex and hippocampus [79]. Recently, in the cortex and hippocampus of rats treated with MPH, MDA levels were increased and SOD levels reduced [81].

The auto-oxidation of catecholamines (dopamine and norepinephrine) can be easily generated and ROS formed [89,90,91]; ROS generation can trigger oxidative damage to DNA and cell death [92,93]. In that sense, it was demonstrated that ATX treatment increases extracellular catecholamine levels [22,72]. Therefore, ATX can trigger an increase of cytosolic and mitochondrial ROS, producing damage to the mitochondria and consequently, cell death [80]. The precise association between the auto-oxidation of catecholamines and the generation of oxidative stress in ADHD remains unclear. Hence, both processes could be implicated in the pathophysiology of ADHD.

## 3. Role of Neuroinflammation

The innate and adaptive immune systems work in harmony to support and determine effective and protective immune responses. The innate immune system works as the first line of defense, including the clearance of microbes such as viruses or bacteria, wound repair, and removal of cells that are in the process of dying. Moreover, the innate immune system can later activate the adaptive system. The cells in the central nervous system that participate in the innate immune response are microglia (which are immune cells in the brain), astrocytes, mast cells, natural killer cells, macrophages and oligodendrocytes, circulating phagocytes, and also monocytes, which are the precursors of macrophages and dendritic cells and play a role in innate immunity [94,95]. The adaptive immune system is highly specific and capable of remembering; it can also effectively initiate responses against previously experienced immunological threats or eliminate tumors. The components of the adaptive immune system are the T and B cells, known as lymphocytes, the effector cells, and their secreted products [94,95].

Inflammation of the nervous system, commonly known as neuroinflammation, can be characterized by the activation of microglia (which play a role in pathological and physiological conditions), astrocytes, oligodendrocytes, and ependymal cells, by increasing levels of proteases, glutamate, ROS, NO, chemokines, toxic cytokines, and prostaglandins and by infiltration of T and B cells, neutrophils, monocytes/macrophages and dendritic cells [96,97,98,99]. The role of neuroinflammation has been associated with several neuropsychiatric disorders, such as autism [100], bipolar disorder [101], depression [102], and schizophrenia [103]. Thus, growing interest points to neuroinflammation as a factor involved in the pathophysiology of ADHD [3,39,40].

Microglia represents the resident immune cells of the CNS, with an important function in the elimination of waste products during inflammation or damage [94]. The cytokine and growth factor S100B is a marker of glial function. In serum samples of children with ADHD, there were no clear differences in the levels of S100B [104]. Conversely, a decreased in the total serum levels of S100B were modestly associated with hyperactive-impulsive symptoms [105].

Astrocytes have unique functional and morphological characteristics that differ within specific areas of the brain, and brain disorders could be characterized by an inflammatory state of the astrocytes. Thus, astrocytes can drive the induction and progression of the inflammatory state, which is notably related to the disorder condition or severity [106]. SynCAM1 is an adhesion molecule involved in synaptic differentiation and organization, which is expressed in astroglial cells. A mice carrying a dominant-negative form of SynCAM1 specifically targeted to astrocytes developed behavioral abnormalities similar to those described in animals model of ADHD, suggesting unappreciated involvement of astrocytes to the pathophysiology of this disorder [107].

An association between cytokines and ADHD symptoms in children has been demonstrated. As a result, elevated levels of IL-16 (hyperactive-impulsive symptoms) and IL-13 (inattention) were found [105]. In patients with ADHD, the adenosine deaminase (ADA) activity, a marker of cellular immunity, was significantly higher; ADA has a role in differentiation and lymphocyte proliferation [55]. Furthermore, it has been suggested that the release of inflammatory cytokines caused by stress or allergic inflammation could alter the maturation of the prefrontal cortex and the neurotransmitters involved in ADHD [108]. Additionally, serum levels of IL-6 were significantly higher in children with ADHD compared with controls [109]. The serum and splenic concentrations of chemokines IP-10, RANTES, and MCP-1 were significantly increased in 5- and 10-week-old SHR, and increased levels of IL-6 and TNF-α were observed in 5-week-old SHR [66]. Recently, in plasma from young people with ADHD, higher levels of C-reactive protein and IL-6 and lower levels of TNF-α and BDNF were found [110].

### 3.1. Inflammation and Polymorphisms

The contribution of gene polymorphisms could be associated as a neurodevelopmental risk factor in the pathogenesis of ADHD. Thus, the findings of the IL-1 receptor antagonist (IL-1RA) gene variable number tandem repeat (VNTR) polymorphism in children with ADHD demonstrated that the 2-repeat allele was associated with reduced risk and the 4-repeat allele with increased risk for ADHD [111]. Nevertheless, no evidence for the association of IL-1RA polymorphism with ADHD was found [112]. Furthermore, a significantly higher polymorphism of dopamine receptor D2 gene (TaqI A) and of BDNF (196 G/A val66met), IL-2 (−330), IL-6 (−174), and TNF-α (−308) was reported [113]. In a study of single nucleotide polymorphisms (SNPs) an association was demonstrated between the cytokine family and the ciliary neurotrophic factor receptor (CNTF) in both adults and children with ADHD, (rs7036351, rs1080750 and rs1124882 risk haplotypes) [114]. Also, two SNPs in the CNTF (rs10758268 and rs7044318) gene were associated with inattentive symptom severity in ADHD and SNPs within cytokine genes IL-16 (rs8039027), and S100B (rs2839361) moderated the association between birthweight and symptom severity [115]. The outcomes of neuroinflammation are summarized in Table 3.

### 3.2. Antibodies in ADHD

A possible association between specific antibodies and immune dysregulation in ADHD has been evaluated and a significant positive immunoreactivity against anti-Purkinje cell antibodies in the cerebellum of children with ADHD was found [117]. Furthermore, a high percentage of anti-Purkinje antibodies and increased serum levels of interleukin IL-6 and IL-10 were detected in patients with ADHD [125]. Moreover, high levels of auto-antibodies against the dopamine transporter [124] and high levels of anti-basal ganglia antibodies were found in ADHD patients [122].

### 3.3. Comorbidity with Other Disorders

A marked comorbidity has been observed between ADHD and asthma [116], atopic eczema [108,118,119], and allergic diseases such as allergic rhinitis, atopic dermatitis, and allergic conjunctivitis [128,129,130,131,132]. Moreover, it was indicated that autoimmune reactions against the basal ganglia and streptococcal infections are more frequent in patients with ADHD [122,123]. The comorbidity of autoimmune diseases with ADHD was demonstrated to be low, but patients with ADHD had a significant prevalence of autoimmune thyroid disease, ulcerative colitis, and ankylosing spondylitis compared to controls [126]. Recently, a large-scale genome-wide cross-trait analysis identified causal links between asthma and ADHD [133]. Thus, more research is required to elucidate the comorbidity between ADHD and allergic or autoimmune disorders.

Both maternal obesity and metabolic complications could increase the risk of ADHD in offspring [120,121]. Moreover, the risk of ADHD in offspring has been found to increase in mothers with inflammatory or immune diseases [38]. Furthermore, a maternal history of autoimmune disease could be associated with an increased risk of ADHD [127], but recently, the maternal C-reactive protein during early pregnancy showed no significant associations with ADHD in offspring [134].

## 4. Use of Dietary and Natural Compounds against Oxidative Stress and Neuroinflammation in ADHD

Increasing studies are looking for alternative therapies for ADHD, mainly focused on the neuroprotective effects of dietary and natural compounds as antioxidants because they may be alternative treatments with fewer side effects. Some nutritional or natural components which have been studied for having therapeutic benefits in ADHD are: Omega-3 fatty acids have antioxidant and anti-inflammatory activities and the two main are docosahexaenoic acid and eicosapentaenoic acid, found mainly in oily fish [3,135,136,137]. *N*-Acetylcysteine is a precursor of the antioxidant glutathione, found in the onion and exerts antioxidant and anti-inflammatory activities [3,135]. Sulforaphane exerts antioxidant and anti-inflammatory activities, is found in highest concentrations in broccoli sprouts, and cauliflower [3]. Ginseng contains a class of phytochemicals called ginsenosides, known as potent antioxidants [2,136]. St. John’s wort, which is rich in flavonoids, providing antioxidants effects [2]. Passionflower contains flavonoids and exerts antioxidants activities [2]. Ginkgo biloba has antioxidant effects and contains flavonoids, terpenoids, and ginkgolic acid [2,135,136]. Several flavonoids with antioxidant activities that include a large group of natural polyphenols are found abundantly in fruits, red wine, green tea, and vegetables [2,135,136]. Thus, it has been shown that these compounds could improve ADHD progression due to their antioxidants and anti-inflammatory properties.

## 5. Conclusions

The pathophysiological process of ADHD has been associated with an increase in oxidative stress and neuroinflammation. Accordingly, some of the factors discussed in this review appear to play a key role in the pathological process of ADHD. Several factors seem to increase oxidative stress, such as the imbalance between oxidants and antioxidants in patients and also the treatment with medications, both of which could increase the oxidative damage in patients. Moreover, several factors can also cause neuroinflammation in ADHD, such as an altered immune response, genetic and environmental associations, comorbidity between ADHD and inflammatory disorders, and also some polymorphisms in inflammatory-related genes. The aforementioned factors offer the potential for dietary and natural compounds as ADHD therapy, due to the potent antioxidant and anti-inflammatory properties such as the increase of antioxidant levels, reduce oxidative stress, and improve the inflammation. In summary, there are several pieces of evidence for the role of oxidative stress and neuroinflammation in the pathophysiology of ADHD. However, clinical trials and prospective, well-designed studies are still needed to confirm these hypotheses.

## Figures and Tables

**Table 1 antioxidants-09-01039-t001:** Summary of oxidative and nitrosative stress biomarkers and outcomes.

Biomarker/Outcome	Sample Compared to Control/Treatment (Tx)	Reference
Improved non-selective attention	Rat intraperitoneal NOS inhibitor	[69]
↑ Extracellular norepinephrine and dopamine in PC	Rat brain-Tx ATX	[22]
Breakdown of PUFAs	↑ exhaled ethane	[49]
↑ Extracellular norepinephrine and dopamine in PC, OC, HPT, HC, and CB	Rat brain-Tx ATX	[72]
NO	↓ Plasma	[70]
↑ TBARS and protein carbonyl formation	Rat brain regions-Tx MPH	[73]
8-oxoG	↑ Plasma	[52]
↑ DNA damage	Rat blood and brain regions-Tx MPH	[74]
↑ Mitochondrial complexes	Rat brain homogenates-Tx MPH	[75]
MDA	↑ Plasma	[50]
MDA	↓ Plasma	[57]
NO	↑ Plasma	[71]
↑ Superoxide in submitochondrial particles in CB and HC	Rat brain-Tx MPH	[76]
MDA and NO	↑ Plasma	[53]
Acrolein-lysine	↑ Urine	[54]
TOS and OSI	↑ Plasma	[58]
MDA and 8-OHdG	↓ Plasma	[56]
XO and NOS	↑ Serum	[55]
↓TBARS and reactive species level in HC and ST ↑ Reactive species level and lipid peroxidation in PC	Rat brain homogenates-Tx MPH	[77]
MDA	↑ Plasma	[51]
↑ TBARS and carbonyl groups	Rat brain homogenates-Tx MPH	[78]
TOS and OSI	↑ Plasma	[59]
TOS	↑ Plasma	[60]
TOS and OSI	↑ Serum	[61]
↑ MDA and induced neurodegeneration in CC and HC	Rat brain homogenates-Tx MPH	[79]
DCFH-DA	↑ Rat brain homogenates	[63]
TOS and OSI	= Serum	[62]
MDA	= Serum	[64]
MDA and 8-OHdG	↑ Plasma and urine	[65]
MDA and free sulphydryl groups	↑ Rat spleen	[66]
Impaired oxidants-antioxidants balance ↑ NO	Serum	[68]
↑ Cytosolic and mitochondrial ROS, damage of mitochondria and cell death	Cell line-Tx ATX	[80]
MDA in CX and HC	↑ Rat brain homogenates-Tx MPH	[81]
Hydroperoxide	↑ Serum	[67]

The table summarizes the oxidative and nitrosative stress biomarkers and outcomes. More details in the text. PC, prefrontal cortex; HC, hippocampus; OC, occipital cortex; CB, cerebellum; ST, striatum; HPT, hypothalamus; CC, cerebral cortex; CX, cortex; ↑, increased; ↓, decreased; =, no difference.

**Table 2 antioxidants-09-01039-t002:** Summary of antioxidant biomarkers and outcomes.

Biomarker/Outcome	Sample Compared to Control/Treatment (Tx)	Reference
TAS	↓ Plasma	[52]
TAS	↓ Plasma	[82]
↑ Adrenaline and noradrenaline ↑ GSSG level and ↓ GSH level	Plasma	[83]
SOD (chronic Tx: ↑ CC, HC, and ↓ ST-acute Tx: ↑ CC and ↓ PC) CAT (acute Tx: ↓ HC)	Rat brain-Tx MPH	[88]
SOD	↓ Plasma	[71]
↑ CAT, ↓ GPx and = SOD	Plasma	[53]
SOD1	↓ Serum	[84]
SOD, GST, GPx, and CAT	↓ Plasma	[85]
Antioxidant activity and CAT	↓ Saliva	[86]
GST, PON1	↓ Serum	[55]
PON1 and thiol	= Plasma	[56]
= Ceruloplasmin and ↑ thiol	Saliva	[87]
TAS	↑ Plasma	[58]
↑ SOD and CAT in CB	Rat brain homogenates-Tx MPH	[77]
PON1 and ARE	↓ Plasma	[51]
↓ SOD and CAT	Rat brain homogenates-Tx MPH	[78]
TAS	↓ Plasma	[59]
↓ TAS and thiol = PON and ARE	Plasma	[60]
TAS, PON1, and ARE	↓ Serum	[61]
↓ GSH, SOD, GPx, and GR in CC and HC	Rat brain homogenates-Tx MPH	[79]
= GSH, SOD, and CAT ↓ GPx in PC	Rat brain homogenates	[63]
↓ Homocysteine and ↑ Folate = Vitamin B12 and TAS	Serum	[62]
Retinyl palmitate and GSH	↑ Plasma and erythrocytes	[65]
TAC, CAT, and GSH	↓ Serum	[64]
Melatonin	↑ Serum	[68]
SOD in CX and HC	↓ Rat brain homogenates-Tx MPH	[81]

The table summarizes the antioxidant biomarkers and outcomes. More details in the text. PC, prefrontal cortex; HC, hippocampus; CB, cerebellum; ST, striatum; CC, cerebral cortex; CX, cortex. ↑, increased; ↓, decreased; =, no difference.

**Table 3 antioxidants-09-01039-t003:** Summary of neuroinflammation and outcomes in ADHD.

Type of Study	Outcome	References
DNA from children	IL-1RA: 2-repeat allele ↓ risk and 4-repeat allele ↑ risk	[111]
DNA from children	No evidence of IL-1RA polymorphism	[112]
DNA from children	↑ Polymorphism of dopamine receptor D2, BDNF, IL-2, IL-6 and TNF-α	[113]
DNA from children and adults	Association with CNTF	[114]
Serum from children	↑ Levels of IL-16 and IL-13 ↓ S100B associated with hyperactive-impulsive symptoms	[105]
A cross-sectional study of adults	↑ Comorbidity with asthma	[116]
Serum from children	↑ ADA activity	[55]
Astrocyte-specific disruption of SynCAM1	ADHD-like behavior abnormalities in mice	[107]
Serum from children	Positive immunoreactivity against anti-Purkinje cell antibodies in the cerebellum	[117]
Birth cohort, population-based and correlational studies of children and adolescents	↑ Comorbidity with atopic eczema	[108,118,119]
DNA from young	2 SNPs in CNTF were associated SNPs within IL-16 and S100B moderated birthweight and symptom severity	[115]
A population-based cohort study using a sibling-comparison design	Maternal obesity and metabolic complications could increase the risk of ADHD in offspring	[120,121]
Serum from patients	Autoimmune reactions against the basal ganglia and streptococcal infections	[122,123]
Serum from children	↑ Auto-antibodies against the dopamine transporter	[124]
Serum from patients	↑ Anti-basal ganglia antibodies	[122]
Serum from children	↑ Anti-Purkinje antibodies and IL-6 and IL-10	[125]
Population-based study of patients	↑ Prevalence of autoimmune thyroid disease, ulcerative colitis, and ankylosing spondylitis	[126]
Population-based nested case-control study	Mothers with inflammatory or immune diseases ↑ risk of ADHD in offspring	[38]
A prospective nationwide study	Maternal history of autoimmune disease could ↑ risk of ADHD	[127]
Population-based case-control, large-scale cross-sectional, population-based studies, and venous blood of children	↑ Comorbidity with allergic diseases such as allergic rhinitis, atopic dermatitis, allergic conjunctivitis	[128,129,130,131,132]
Serum from children	↑ IL-6	[109]
Serum and spleen from SHR	↑ IP-10, RANTES, and MCP-1 ↑ Levels of IL-6 and TNF-α	[66]
Large-scale genome-wide cross-trait association study	Causal links between asthma and ADHD	[133]
Plasma from young	↑ C-reactive protein and IL-6 and ↓ TNF-α and BDNF	[110]
Prenatal studies with a nested case-control design	Maternal C-reactive protein during early pregnancy showed no significant association in offspring	[134]

The table summarizes the neuroinflammation and outcomes. More details in the text. ↑, increased; ↓, decreased.

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
