# Peer review of "Role of Oxidative Stress and Neuroinflammation in Attention-Deficit/Hyperactivity Disorder"

_antioxidants, 2020, doi:10.3390/antiox9111039_

Round 1
Reviewer 1 Report
This paper by J.C. Corona on the role of oxidative stress and neuroinflammation in ADHD is a short, straightforward and effective review that leads the reader to understand the current knowledge on this complex matter. By focusing on the disorder, the role of oxidative stress and iii) neuroinflammation on the disorder the Author covers much of the questions known on the subject. About ¼ of the cited literature refer to 2018-to-2020 publications, which makes this review rather up-to-date.
Minor points
- Page 4, line 133 (2.3 Antodixant levels in ADHD). ‘… (Ceylan, M., 2010). …’ is ‘… [50]. …’.
- Page 7, line 231 (Table 3). As the number of genes potentially involved in ADHD is steadily increasing, it could extremely useful for the reader if the Author would add a column to Table 3 with the indication of the type and localization of the polymorphisms along the gene/protein sequences if/when relevant for the outcome.
- Page 9, line 256 (4. Conclusion). There are some recent evidences claiming the effect of certain strictly dietary components on ADHD. It could be worth for the reader that the Author makes a very brief reference to this aspect of ADHD research in this conclusion section.
Author Response
Response to reviewer number 1:
Minor points
1) Page 4, line 133 (2.3 Antodixant levels in ADHD). ‘… (Ceylan, M., 2010). …’ is ‘… [50]. …’.
Response: I have corrected the reference.
2) Page 7, line 231 (Table 3). As the number of genes potentially involved in ADHD is steadily increasing, it could extremely useful for the reader if the Author would add a column to Table 3 with the indication of the type and localization of the polymorphisms along the gene/protein sequences if/when relevant for the outcome.
Response: By suggestion of some reviewers who consider Table 3 too large, I will not add a new column. However, I added in the text what suggested by the reviewer regarding the polymorphisms.
3) Page 9, line 256 (4. Conclusion). There are some recent evidences claiming the effect of certain strictly dietary components on ADHD. It could be worth for the reader that the Author makes a very brief reference to this aspect of ADHD research in this conclusion section.
Response: I have included a new section including the use of dietary components and natural antioxidants which can improve the oxidative stress and neuroinflammation in ADHD.
Reviewer 2 Report
The author performs a thorough review of the different factors that increase
oxidative stress and neuroinflammation and might be related with ADHD.
One of my suggestions is related to use of antioxidants that could have potential as a treatment against oxidative stress and neuroinflammation in ADHD. I think that the manuscript would improve if a paragraph was included discussing about this topic. Some useful references might be:
- Joshi K, Lad S, Kale M, Patwardhan B, Mahadik SP, Patni B, Chaudhary A, Bhave S, Pandit A. Supplementation with flax oil and vitamin C improves the outcome of Attention Deficit Hyperactivity Disorder (ADHD). Prostaglandins, Leukotrienes and Essential Fatty Acids. 2006 Jan 1;74(1):17-21.
- Verlaet AA, Maasakkers CM, Hermans N, Savelkoul HF. Rationale for dietary antioxidant treatment of ADHD. Nutrients. 2018 Apr;10(4):405.
- Richardson AJ. Omega-3 fatty acids in ADHD and related neurodevelopmental disorders. International review of psychiatry. 2006 Jan 1;18(2):155-72.
- Moghadas M, Essa MM, Ba-Omar T, Al-Shehi A, Qoronfleh MW, Eltayeb EA, Guillemin GJ, Manivasagam T, Justin-Thenmozhi A, Al-Bulushi BS, Al-Adawi S. Antioxidant therapies in attention deficit hyperactivity disorder. Frontiers in bioscience (Landmark edition). 2019 Jan;24:313-33.
Another suggestion is to improve the tables, they are large and hard to follow.
Finally, I find the conclusion very short and underdeveloped. At its current state it is just a very short summary of the paper.
Author Response
Response to reviewer number 2:
1) One of my suggestions is related to use of antioxidants that could have potential as a treatment against. I think that the manuscript would improve if a paragraph was included discussing about this topic.
Response: I have included a new section including the use of natural antioxidant and dietary components which can improve the oxidative stress and neuroinflammation in ADHD.
2) Another suggestion is to improve the tables, they are large and hard to follow.
Response: I have corrected the Tables a little bit. I hope that with the changes they are now easy to follow.
3) Finally, I find the conclusion very short and underdeveloped. At its current state it is just a very short summary of the paper.
Response: I have modified the conclusion and I hope the modifications are to the liking of the reviewer.
Reviewer 3 Report
Antioxidants 896881 –
This review paper focuses on oxidative stress and neuro-inflammation in ADHD. I found the content and the writing to be generally quite good. I am an experimental psychologist and have done quite a bit of ADHD research in the past. Also, I did study neurobiology as a graduate student, but don’t consider myself an expert on the main topics of the paper. I am also unfamiliar with most of the papers in the reference list. Hopefully, one of the other reviewers can provide more in depth comments on oxidative stress and (neuro)inflammation.
General comments:
1. There is not much specificity with respect to symptom clusters in the oxidative/antioxidant sections. It is important to know whether findings are specific to one subtype or another. Also, it would be important to know whether there is any information for how subtypes change across the course of development. There are clearly more mentions to symptom clusters in the inflammation section(s).
2. I do think the paper would benefit from more explicit links between oxidative stress and inflammation and pathological / pathophysiological processes. Multi-factorial models typically proceed from genes to gene × environment interactions to brain structure / function to cognitive phenotypes. I’d be interested in which of these different “levels” are assumed or shown to have interactions with oxidative stress and neuroinflammation.
3. Lines 156-158: I noticed that none of the “conflicting” results been summarized/presented in the text. Instead, only positive results or the authors interpretation of positive results. I think this point is important, if there are negative findings (or areas of clear mixed findings) in the literature that those be highlighted. Possibly put the opposing results (cf) in the Tables (i.e. right hand column) for example [49,cf.50]. In short, the controversy should be highlighted rather than glossed over in a review paper.
Minor Comments:
1. Line 9: ADHD affects people throughout the lifespan.
2. Line 37: “more frequent”
3. Line 133: reference not numbered (Ceylan, 2010).
Author Response
Response to reviewer number 3:
General comments:
1) There is not much specificity with respect to symptom clusters in the oxidative/antioxidant sections. It is important to know whether findings are specific to one subtype or another. Also, it would be important to know whether there is any information for how subtypes change across the course of development. There are clearly more mentions to symptom clusters in the inflammation section(s).
Response: In some papers, there is a description of the different subtypes used indistinctly (inattentive, hyperactive/impulsive, combined or no specified) but there is no mention or association with the outcome obtained. Neither, there is no description in other papers respect to symptom clusters or about one subtype or another in oxidative/antioxidant sections. Therefore, it was only described in the sections where there was a clear description of symptoms clusters as in the case of the inflammation section.
I agree with the reviewer that it would be important to know whether findings are specific to one subtype or another and if these are related to a decrease or increase in the oxidative stress.
2) I do think the paper would benefit from more explicit links between oxidative stress and inflammation and pathological / pathophysiological processes. Multi-factorial models typically proceed from genes to gene × environment interactions to brain structure / function to cognitive phenotypes. I’d be interested in which of these different “levels” are assumed or shown to have interactions with oxidative stress and neuroinflammation.
Response: I thank the reviewer for this helpful comment. I have added some paragraphs about the interactions between oxidative stress and inflammation (pages 2-3).
3) Lines 156-158: I noticed that none of the “conflicting” results been summarized/presented in the text. Instead, only positive results or the authors interpretation of positive results. I think this point is important, if there are negative findings (or areas of clear mixed findings) in the literature that those be highlighted. Possibly put the opposing results (cf) in the Tables (i.e. right hand column) for example [49,cf.50]. In short, the controversy should be highlighted rather than glossed over in a review paper.
Response: I thank the reviewer for this helpful comment. In the text and the tables, the results are specified, whether positive or negative, or failing that if there were no changes, either with downward or upward arrows or with the = sign. Therefore, the mentioned paragraph was added to specify that there are some opposite results "Some of the opposing results observed in the different studies could be explained by the different methodologies, the participant selection criteria and the analysis used."
Minor Comments:
1) Line 9: ADHD affects people throughout the lifespan.
Response: Is now corrected.
2) Line 37: “more frequent”
Response: Is now corrected.
3) Line 133: reference not numbered (Ceylan, 2010).
Response: reference is now numbered.
Reviewer 4 Report
This manuscript provided a brief review regarding the roles of oxidative stress and neuroinflammation in ADHD. The whole manuscript was divided into 4 sections, Introduction, Role of oxidative stress, Role of neuroinflammation, and Conclusion. The major concerns of this manuscript are the massive references without main conclusion. For example, 2.1. Oxidative stress and oxidant levels. The single paragraph was written only mentioning relevant references from Reference 47-64. A vast array of references independent on their pro and con were cited and findings were described, one by one. The description should be organized as main flow and aims, not just saying. The same concerns were also to other sections.
Author Response
Response to reviewer number 4:
1) The major concerns of this manuscript are the massive references without main conclusion.
For example, 2.1. Oxidative stress and oxidant levels. The single paragraph was written only mentioning relevant references from Reference 47-64. A vast array of references independent on their pro and con were cited and findings were described, one by one. The description should be organized as main flow and aims, not just saying. The same concerns were also to other sections.
Response: I thank the reviewer for this helpful comment. The main aim of this review is to present a general outlook about the factors that increase oxidative stress and neuroinflammation in ADHD.
Oxidative stress and neuroinflammation play a role in the pathophysiology of ADHD due to the imbalance between oxidants and antioxidants, catecholaminergic dysregulation, medications used for treatment, genetic and environmental factors, all those factors could be producing oxidative stress and neuroinflammation, which further increases the symptoms and as a result triggering a vicious circle.
Therefore, I include the references with the pros and cons (2.1. Oxidative stress and oxidant levels), first following a flow of these factors that produce oxidative stress; secondly, medications used for treatment and their relation with oxidative stress and later the factors that produce neuroinflammation.
Round 2
Reviewer 3 Report
My issues were addressed.
Author Response
The issues were addressed
Reviewer 4 Report
A vast array of references were described and cited in all the manuscript. However, repeated and controversial findings were not summarized and discussed. The follow is an example. Can the author categorize, compare, and conclude the main findings.
2.1. Oxidative stress and oxidant levels
Numerous studies have demonstrated elevated levels of oxidative stress in ADHD, increasing the evidence for oxidative stress being a pathophysiological factor. The measurement of higher rates of ethane levels, as a non-invasive measure of oxidative breakdown of n-3 PUFAs in patients with ADHD, was also demonstrated [49]. Furthermore, elevated levels of malondialdehyde (MDA) have been observed in children with ADHD [50,51]. A randomized, double-blind, placebo-controlled study found that children with ADHD had increased damage to DNA, measured with 8-oxo-7,8-dihydroguanine (8-OxoG) [52]. Moreover, the levels of MDA in plasma from children and adolescents with ADHD were significantly higher than the controls [53]. An increase in lipid peroxidation in paediatric ADHD patients was evaluated using acrolein–lysine in urine samples [54]. Also, changes in plasma levels of xanthine oxidase (XO) were significantly higher in patients with ADHD [55]. The levels of MDA and the DNA damage indicator 8-hydroxy-2′-deoxyguanosine (8-OHdG) were statistically lower in children with ADHD [56,57]. The total oxidative status (TOS) and oxidative stress index (OSI) were higher in patients with ADHD than controls [58]. Furthermore, in children and adolescents with ADHD the TOS and OSI were significantly higher than in healthy controls [59]. Also, the TOS was high in plasma of children and adolescents with ADHD [60]. In a meta-analysis of ADHD patients, an increase in oxidative stress was found [37]. Moreover, the TOS and OSI were increased in children with ADHD [61]. No significant differences were detected in serum TOS and OSI levels in adults with ADHD [62]. Also, in the spontaneously hypertensive rat (SHR), used as an animal model for ADHD, an increase in ROS production measured using 2′-7′-dichlorofluorescein diacetate (DCFH-DA) was demonstrated in the striatum, hippocampus and cortex [63]. In contrast, it was demonstrated that MDA levels were not significantly different in children with ADHD [64]. Moreover, in children with ADHD increases in plasma MDA and urinary 8-OHdG levels were found [65]. Finally, the levels of MDA and free sulphydryl groups in the spleen were higher in 5-week-old SHR than in control rats [66]. Recently, evaluation of serum levels of hydroperoxide, an oxidative stress marker, was shown to be higher in preschool children with ADHD [67].
Author Response
Response: I hope now that I have adequately answered to the valuable comments. I have added some paragraphs about the controversial findings, a discussion and the conclusion about the main findings (page 6, line 182 to 194).
Round 3
Reviewer 4 Report
There was no additional comments.